# Variables Determining Higher Home Care Effectiveness in Patients with Chronic Cardiovascular Disease

**DOI:** 10.3390/ijerph19095170

**Published:** 2022-04-24

**Authors:** Elżbieta Szlenk-Czyczerska, Marika Guzek, Dorota Emilia Bielska, Anna Ławnik, Piotr Polański, Donata Kurpas

**Affiliations:** 1Department of Health Sciences, University of Opole, 68 Katowicka Street, 45-060 Opole, Poland; 2Medical and Diagnostic Centre (MDC), 9 Niklowa Street, 08-110 Siedlce, Poland; marika.guzek@centrum.med.pl; 3Department of Family Medicine, Medical University of Białystok, 1 J. Kilińskiego Street, 15-089 Białystok, Poland; d.bielska1@wp.pl; 4Faculty of Health Sciences, John Paul II University of Applied Sciences in Biala Podlaska, 95/97 Sidorska Street, 21-500 Biala Podlaska, Poland; lawnikania@gmail.com; 5Family Physician’s Practice, Non-Public Healthcare Center, 4 Nad Potokiem Street, 58-350 Mieroszow, Poland; p.polanski@wp.pl; 6Department of Family Medicine, Wrocław Medical University, 1 Syrokomli Street, 51-141 Wrocław, Poland; dkurpas@hotmail.com

**Keywords:** cardiovascular diseases, home care services, patients, caregivers

## Abstract

The aim of this cross-sectional study was to analyze the variables that influence the effectiveness of home care in patients with chronic cardiovascular disease and their informal caregivers. The study was conducted in 193 patients and their 161 informal caregivers. The study used the WHOQOL-BREF Quality of Life Questionnaire, the health behavior inventory questionnaire (HBI), the Camberwell assessment of need short appraisal schedule (CANSAS) and the hospital anxiety and depression scale–modified (HADS–M) version. Spearman’s rank correlation coefficient test and logistic regression were used for analyses. Analysis of patients revealed an association between home care effectiveness and the following variables (OR per unit): age (OR = 0.98, 95% CI: 0.95–0.99), educational level (OR = 1.45, 95% CI: 1.05–2.02), financial status (OR = 0.43, 95% CI: 0.21–0.83), medication irregularity (OR = 0.25, 95% CI: 0.07–0.72), presence of comorbidities (OR = 6.18, 95% CI: 1.83–23.78), health care services provided by a nurse (OR = 1.25, 95% CI: 1.03–1.64), and number of visits to a cardiology clinic (OR = 1.25, 95% CI: 1.02–1.59). There was no association between care effectiveness and sex (*p* = 0.28), place of residence (*p* = 0.757), duration of cardiovascular disease (*p* = 0.718), number of home visits (*p* = 0.154), nursing interventions (*p* = 0.16), and adherence to lifestyle change recommendations (*p* = 0.539) or proper dietary habits (*p* = 0.355). A greater chance of improved health care effectiveness was found in patients whose caregivers reported higher social (OR = 1.24, 95% CI: 1.09–1.44), psychological (OR = 1.68, 95% CI: 1.25–2.37), and physical (OR = 1.24, 95% CI: 1.05–1.49) quality of life. Patients with cardiovascular disease who were characterized by lower educational attainment, poorer financial status, fewer visits to cardiology clinics, lower utilization of medical services, poorer self-perception of mental and physical well-being, recent onset of disease symptoms, and irregular use of medications, were much more likely to have poorer health care effectiveness. Patients with cardiovascular disease and their caregivers can be well supported at home as long as the care model is tailored to the specific needs. This includes family care coordination in the health care team, home care, and general practice support.

## 1. Introduction

Cardiovascular disease (CVD) is a major cause of death and disability both globally [1,2] and in the EU [3], and represents a significant burden on healthcare systems and national budgets [1,2,3]. Therefore, improving health care for these patients seems to have become absolutely necessary [4].

There are many unresolved problems in CVD patients [1,2,3]. Most patients with chronic diseases (including cardiovascular diseases) have an increased need for medical care [5]. Promoting the enhancement of protective factors (disease surveillance and support through social health programs) and the adoption of healthy behaviors are important strategies to reduce the burden of noncommunicable diseases [6]. It encourages various countries to improve their primary health care (PHC) to increase its effectiveness and reduce health inequities [7]. However, despite significant improvements, there is still a large gap between patients’ needs and the quality and effectiveness of health care [8]. Therefore, all current activities aim to introduce models that focus on the patient with chronic diseases [6,9], with particular attention to the home environment [10]. Adequate home care organized within the health care system can meet the needs of today’s patients and their caregivers [7,11,12] and contribute to a decrease in the number of people suffering from chronic diseases [11,13,14]. Home care has been shown to have high clinical effectiveness [15], reduce the risk of hospitalization, increase satisfaction, and reduce costs compared with hospital care [10,16], and reduce the burden on caregivers and improve their quality of life [17]. Other studies show that implementation of a regional health program for patients with chronic heart failure based on a chronic care model strategy delivered in the primary care setting ultimately resulted in a lower risk of hospitalization for heart failure (34%) and improved survival (18%). As observed, the impact on hospitalization was mainly due to a 50% higher rate of planned hospitalizations [18].

Strengthening health care is a key strategy to alleviate the increasing burden of cardiovascular disease [10]. Expected disease morbidity (the most common diagnosis in patients over 18 years of age in 2019 was cardiovascular disease −2.8%) and demographic changes will pose a major challenge to health care delivery [19]. As a result of these changes, home care in PHC remains a service with great potential and is still evolving [16]. Identifying the variables that impact improving the effectiveness of PHC health care and combining them with known professional initiatives can help achieve the desired outcomes thanks to the relevance of the data [8].

Given the role of PHC in professional and interdisciplinary health care for CVD patients, it is critical to determine a targeted and most effective model for home care. Improving health care delivery by PHC teams will effectively impact the functioning of patients and their caregivers and reduce the cost of care and unnecessary hospitalizations [10]. However, as far as we know, there are few studies investigating the effectiveness of home-based care in patients with CVD [20,21,22]. Although there are publications, they focus on clinically homogeneous groups (e.g., heart failure or stroke) [17,23,24].

With this in mind, this study aims to analyze the variables that influence the effectiveness of home care in patients with chronic cardiovascular disease and their informal caregivers.

## 2. Materials and Methods

### 2.1. Study Design and Setting

The present study has a cross-sectional and observational design and is part of a larger study aimed at identifying indicators that determine the effectiveness of home care for patients with CVD and variables that determine effective support systems for their home caregivers [20,21,22]. The research was conducted between 2016 and 2017 after obtaining approval from the Bioethics Committee of the Medical College in Wroclaw (KB-86/2016) in compliance with the requirements of the Declaration of Helsinki of 1975 (amended in 2000). The study was conducted among Polish patients with CVD and their informal caregivers. These patients were cared for by family caregivers working in primary care. Eight primary care centers in Opole, Lubelskie, Mazowieckie, Dolnośląskie, and Podlaskie voivodships participated in the study.

### 2.2. Participants

Invitations to the study were sent to the heads of ten primary health care centers in Opole, Lubelskie, Mazovia, Dolnośląskie, and Podlaskie voivodeships. Eight of them responded positively, so only patients from these centers were included in the study. The family nurse practitioner identified patients for the study according to the inclusion criteria. Chronic CVD was defined based on the primary care history (primary care physician diagnosis). Two hundred patients with CVD who were cared for at home by family members and one hundred eighty informal caregivers were invited to participate in the study. The final sample of participants was determined based on their time availability. Ultimately, 193 (97%) patients participated in the survey and 161 (89%) of their informal caregivers. Patients and their caregivers were encouraged to participate in the study by a family nurse practitioner during scheduled home visits. The family nurse practitioner conducted an interview with the patient and his or her informal caregiver in which she introduced the purpose and method of the study and obtained preliminary verbal consent. Both patients and caregivers received a set of questionnaires and a written informed consent form to participate in the study. Caregivers and nurses completed an additional questionnaire about the patient (i.e., paired questionnaires about the same patient). For patients, the following criteria for participation in the study applied: they had to be at least 18 years old, have received a CVD diagnosis (ICD-10 codes) at least 12 months before the study, and live at home under the care of a family caregiver. For informal caregivers, the following criteria for participation in the study applied: at least 18 years of age and caring for a patient with chronic cardiovascular disease outside the home care setting for at least 12 months before the study. The exclusion criteria (disqualification by family caregivers) in both groups were cognitive impairment and other severe mental illness and/or other difficulties preventing active participation in the study. The final number of participants was based on their time availability. Ultimately, 193 patients and 161 informal caregivers participated in the survey. The sample group selection is shown in Figure 1.

Participation in the study was voluntary and anonymous. Before the study, each patient and his or her caregiver were informed of the aim, the method, and the possibility of withdrawal at each stage of the study. The aim and procedures were explained during the selection phase, and only those who voluntarily consented were accepted. Patients and their caregivers completed the questionnaire in person (paper–pencil method), with a family nurse present.

### 2.3. Variables and Data Collection 

#### 2.3.1. The WHOQOL-BREF Quality of Life Questionnaire 

The patient’s quality of life was assessed using the short version of the World Health Organization Quality of Life Questionnaire (WHOQOL) [25]. This questionnaire assesses quality of life in four main domains, such as physical, psychological, social relationships, and environment. The test also includes some separately scored questions on individual perception of quality of life (question 1) and health status (question 2). Responses were recorded on a 5-point Likert scale. The reliability of the Polish version of WHOQOL-BREF was checked with the α-Cronbach coefficient, which was 0.81 for physicality, 0.78 for psychology, 0.69 for social relations, and 0.77 for environment. Internal consistency for the whole questionnaire was 0.90 [26,27]. 

#### 2.3.2. The Health Behavior Inventory Questionnaire

To assess the degree of health promoting behavior in CVD patients, Juczyński’s health behavior inventory questionnaire (HBI) was used. It consists of 24 statements that assess four categories of health-promoting behaviors, including proper eating habits, prophylactic activities, and appropriate health attitudes and practices. Respondents rate each statement on a scale in which 1 represents almost never, 2 represents rarely, 3 represents sometimes, 4 represents often, and 5 represents most/almost always. The scores are then added together to calculate the overall intensity of health activities, which ranges from 24 to 120 points. The higher the score, the higher the intensity of the health-promoting behavior. In addition, the intensity of each category is assessed separately [28].

#### 2.3.3. The Camberwell Assessment of Need Short Appraisal Schedule 

The extent to which needs were met or not met was assessed using a modified version of the Camberwell modified short needs assessment (CAN). It was developed by focus groups and expert judges and used to assess the needs of patients in the emergency department and primary care physicians. The modification of CAN covers 22 problem areas, and the Camberwell need index is calculated for the study. Calculations include determining the number (N) of met (1) and unmet (0) needs of patients using 24 questions covering 22 needs. After the number (N) of needs of a respondent is determined, the number (M) of met needs (1) is determined. The formula M/N is used to calculate the Camberwell index. In addition, the Camberwell index for the number of unmet needs is also accessible via the formula 1-M/N (not used in this analysis). The α-Cronbach coefficient for the modified version of CAN was 0.82 [13].

#### 2.3.4. The Hospital Anxiety and Depression Scale-Modified Version 

The modified version of the hospital anxiety depression scale (HADS) questionnaire was used to assess anxiety and depression (HADS–M). The instrument uses seven items to measure anxiety, seven to assess the degree of depression, and two more to assess nervousness and aggression. The questionnaire is suitable for assessing anxiety, depression, and aggression in both inpatients and outpatients. It contains 16 test questions that are scored from 0–3. The score is the sum of the points in each category. The maximum score for anxiety or depression is 21, and for aggression is 6. The first two subscales are interpreted as follows: 0–7—normal behavior, 8–10—borderline mild anxiety, 11–21—pathological and indicative of an anxiety syndrome. Validation studies of the original and modified versions of the HADS demonstrate their reliability and accuracy. The Spearman correlation coefficient between the test items and the total score in each subscale was statistically significant (at least *p* ˂ 0.01) and ranged from 0.41 to 0.76. The accuracy of the test was assessed by comparing the HADS scale scores with the interview scores. The correlation coefficient for the anxiety subscale was 0.54 and for the depression subscale—0.79 [29,30].

#### 2.3.5. Authors’ Self-Prepared Questionnaire 

Sociodemographic data of CVD patients were collected using a questionnaire designed by the authors themselves, which included data such as age, sex, marital status, educational level, financial status, place of residence, and expectations of a PHC physician/nurse. In addition, the questionnaire included information on the number of visits to GP, the number of visits to the cardiology clinic, the type of services and interventions provided by a nurse, and the number of interventions and home visits performed by a GP/nurse in the past 12 months. 

### 2.4. Ethical Aspects

The study was approved by the Bioethical Committee at Medical University in Wroclaw (no. KB-86/2016).

### 2.5. Statistical Methods 

The results of the study were statistically analyzed using the statistical package R (version 3.4.0).

For quantitative variables, the arithmetic mean, standard deviation, first quartile (Q.25%), median (Q.50%), third quartile (Q.75%), minimum, and maximum were calculated. For nominal variables, the frequency (i.e., percentage) was determined. The Shapiro–Wilk test showed that only a few variables had a standard normal distribution, namely WHOQOL-BREF in the areas of body, mind, and environment. The distributions of the other variables deviated completely from a standard normal distribution. Therefore, the chi-square test, Fisher’s test, Wilcoxon’s test, and Spearman’s rank correlation coefficient were used for further analysis. The test probability at the level of *p* ≤ 0.05 was considered significant. The null hypothesis (H0) was rejected when the *p* value < 0.05 (α = 0.05).

To define higher (HEHC) and lower (LEHC) health care effectiveness and to examine the differences between them, a criterion was established based on total scores for quality of life (WHOQOL-BREF), health promoting behaviors (HBI), and need satisfaction (Camberwell Index).

The patient was included in the HEHC group if his scores for WHOQOL-BREF), HBI, and Camberwell index were higher than the corresponding values of 25% of the quantiles (first quartiles) of these three variables. Otherwise, the patient was included in the LEHC group (Figure 2). The analogous criterion in patients cared for by a nurse at home based on the median (50% quantile) had a disadvantage: the size of one group was three times that of the other. For statistical reasons, both compared groups should have a similar size.

Respondents were divided into two groups: LEHC and HEHC. The patient was included in the HEHC group if all scores (Camberwell, WHOQOL-BREF, and HBI) were above their 25% quantiles. The other patients were included in the LEHC group. This criterion allowed the formation of two groups that were comparable in size. Based on this criterion, a new variable was defined, effective health care (EHC), characterized by the following values:

0—if a patient belongs to a group with worse effectiveness of health care.

1—if a patient belongs to a group with better effectiveness of health care.

For the analysis of logistic regression, only the explanatory variables that correlated with EHC (an explained variable) at a significance level of 0.05 were selected. The analysis yielded 22 such variables. Then, all possible models created from the 22 subsets of explanatory variables and the explained variable were examined. The analysis was performed only for the models in which all variables were statistically significant at the 0.05 level.

The models that consisted of 8 explanatory variables (16 models defined by a total of 17 explanatory variables) were the strongest. Finally, four models consisting of 17 explanatory variables were selected. To investigate the remaining 5 explanatory variables, 2 more (consisting of 7 explanatory variables-including 4 of the 5 missing variables) were added. The variable ‘diagnosis ICD-10: I69′ did not appear in any model. After a more detailed analysis, its correlation with EHC was found to be atypical. 

To describe the correlation of EHC with the variables examined in the study, logistic regression analysis was performed. It allowed the description of the correlations based on the odds ratio.

In describing the differences between the LEHC and HEHC groups, we usually use the values of the Me-medians. In cases where both medians are the same, we also report the mean in parentheses, for example, LEHC: 0 (0.39) vs. HEHC: 0 (1.14).

## 3. Results

### 3.1. Sociodemographic Data of Patients with CVD

To verify the differences between patients with LEHC and HEHC cared for at home, an analysis of sociodemographic data were performed. It was found that respondents with LEHC were older than those with HEHC (LEHC: Me—77.5 vs. HEHC: Me—70; *p* = 0.032). Patients with LEHC were mostly characterized by primary (40.5%, *n* = 34) and vocational (26.2%, *n* = 22) educational attainment. In addition, fewer patients in this group had secondary education (without high school diploma—21.4%, *n* = 18 and with high school diploma—4.8%, *n* = 4) or higher education (4.8%, *n* = 4) (*p* = 0.025). It was also found that the studied patients with LEHC were characterized by a smaller group of patients with very good (1.2%, *n* = 1) or good (15.9%, *n* = 13) financial status, and more patients reported average (57.3%, *n* = 47) or bad financial status (25.6%, *n* = 21) (*p* = 0.004). No statistically significant differences were found for the variables relationship status (*p* = 0.085) and place of residence (*p* = 0.757). Compared to individuals with HEHC (*p* = 0.007), more individuals with LEHC received social services (21.4%, *n* = 18) (Appendix A).

### 3.2. Clinical Data of CVD Patients with Worse (*n* = 84) and Better (*n* = 85) Health Care Effectiveness

Data on the clinical status of the respondents are presented below. No statistically significant differences were found between respondents with LEHC or HEHC and CVD duration, number of home visits, telephone consultations, nursing interventions, health care services provided by a nurse in the past 12 months, current CVD treatment status (data collected by a nurse practitioner), or adherence to lifestyle change recommendations and proper dietary habits. However, the group with LEHC reported fewer cardiology clinic visits within the past 12 months compared with patients with HEHC (LEHC: Me—0.5 vs HEHC: Me—2, *p* = 0.007). Analysis of the type and number of health care services in patients with LEHC and HEHC revealed that patients with LEHC used other health care services less frequently than patients with HEHC (Me: 0 (0.39) vs. Me: 0 (1.14), *p* = 0.032). The group of investigators in home care with LEHC was characterized by a lower number of patients who rated their physical well-being as good (LEHC: 22.6% vs. HEHC: 52.9%, *p* < 0.001). Mental status assessment showed that the number of patients with LEHC who reported their mental well-being as quite good is equal to (LEHC: 24.1% vs. HEHC: 21.4%), as good (LEHC: 24.1% vs. HEHC: 59.5%) or very good (LEHC: 2.4% vs. HEHC: 10.7%) less than, and as bad (LEHC: 42.2% vs. HEHC: 8.3%) or very bad (LEHC: 7.2% vs HEHC: 0%) much more than that of patients with HEHC (*p* < 0.001). Significant differences were found between the level of effectiveness of medical care and current symptoms or regular use of prescribed medications. Patients with LEHC suffered more frequently from some CVD-related symptoms (LEHC: 63.1% vs. HEHC: 42.9%, *p* = 0.011) and took less systematically prescribed medications (LEHC: 64.3% vs. HEHC: 85.9%, *p* = 0.001) than patients with HEHC (Table 1, Appendix A).

### 3.3. Differences between Caregivers of Patients with Worse (*n* = 77) and Better (*n* = 57) Health Outcomes

No significant differences were found between caregivers of patients with LEHC or HEHC and sociodemographic data (Appendix A). Analysis showed that caregivers of patients with LEHC compared to caregivers of patients with HEHC had lower scores in physical (Me: 13.14 vs. 14.29, *p* = 0.035), psychological (Me: 12.67 vs. 14, *p* = 0.003) and social relationships (Me: 14.67 vs. 16, *p* = 0.044), and lower intensity of health-promoting behaviors in the positive mental attitude category (Me: 3.5 vs. 3.75, *p* = 0.036) (Appendix A).

### 3.4. Logistic Regression Analysis and Odds Ratio for the Effectiveness of Health Care in the Group of Patients and Their Caregivers

The logistic regression analysis and odds ratio in the group of patients and their caregivers led to the selection of models with eight explanatory variables that allowed the calculation of the odds ratio for EHC.

#### 3.4.1. Logistic Regression Analysis and Odds Ratio—Model 1 (*n* = 130)

In patients differing in age by 77 years, younger patients were 6.76 (OR = 0.15, 95% CI: 0.02–0.88) times more likely to have better EHC than older patients. In patients who differ in age by 1 year, this chance decreases to 1.03 times (OR = 0.98, 95% CI: 0.95–0.99) in favor of younger patients.

Patients who are in a good financial position have a 12.26 (OR = 0.08, 95% CI: 0.01–0.58)-fold higher odds of better EHC than patients whose financial position is poor. However, patients with better financial situation who differ by only 1 point on the 5-point scale on this question have a 2.31 (OR = 0.43, 95% CI: 0.21–0.83)-fold higher odds.

Patients with high scores on the HADS–M aggression scale were 6.42 (OR = 6.42, 95% CI: 1.33–35.23) times more likely to have better EHC than patients who were not aggressive at all. However, those with higher aggression scores, differing by only 1 point on the HADS–M scale, have a 1.36 (OR = 1.36, 95% CI: 1.05–1.81)-fold higher risk.

Those patients whose caregivers report strong social relationships (20 on the WHOQOL-BREF scale) have 32.62 (OR = 32.62, 95% CI: 4.13–333.84) times greater odds of better EHC than those whose relationships are weaker (4 on the WHOQOL-BREF scale). However, those whose caregivers score 1 point higher report a 1.24 (OR = 1.24, 95% CI: 1.09–1.44) times greater chance (Table 2 and Appendix A).

#### 3.4.2. Logistic Regression Analysis and Odds Ratio—Model 2 (*n* = 130)

Those with postgraduate education have 9.12 (OR = 9.12, 95% CI: 1.37–68.87) times higher odds of better EHC than those with primary education. However, those who differ by one position in their favor on the seven-point education scale have 1.45 (OR = 1.45, 95% CI: 1.05–2.02) times higher odds than those with lower levels of education (Table 2 and Appendix A).

#### 3.4.3. Logistic Regression Analysis and Odds Ratio—Model 3 (*n* = 120)

The patients whose caregivers report high scores in the psychological domain of quality of life (18 on the scale WHOQOL-BREF) have 89.55 (OR = 89.55, 95% CI: 6.89–1768.19) times higher chance of better quality of life than those whose scores are lower (9.33 on the scale WHOQOL-BREF). The patients whose caregivers differ by 1 point in their favor in this category have 1.68 (OR = 1.68, 95% CI: 1.25–2.37) times higher odds (Table 2 and Appendix A).

#### 3.4.4. Logistic Regression Analysis and Odds Ratio—Model 4 (*n* = 124)

Those patients who visited a cardiology clinic 24 times in the past 12 months were 198.49 (OR = 198.49, 95% CI: 1.63–70,383.3) times more likely to have better EHC than those who did not visit a clinic in that time. Those who differed by one visit in their favor had 1.25 (OR = 1.25, 95% CI: 1.02–1.59)-fold higher odds (Table 2 and Appendix A).

#### 3.4.5. Logistic Regression Analysis and Odds Ratio—Model 5 (*n* = 124)

Those patients who rated their mental status as very good had 7.55 (OR = 7.55, 95% CI: 1.19–56.44) times higher odds of better EHC than those who reported low scores on this item. Those who score 1 point in their favor on this category have a 1.66 (OR = 1.66, 95% CI: 1.04–2.74) times greater chance (Table 2 and Appendix A).

#### 3.4.6. Logistic Regression Analysis and Odds Ratio—Model 6 (*n* = 123)

The patients who were not diagnosed with any urologic disease (as reported by the nurse) had 6.20 (OR = 0.16, 95% CI: 0.04–0.55) times higher odds of better EHC than the patients who were diagnosed with some urologic diseases (Table 2 and Appendix A).

## 4. Discussion

### 4.1. Main Results

The analysis included the variables that could influence the level of EHC. Those with higher EHC were younger, better educated, and had better financial status than those with potential LEHC. Clinical factors associated with EHC included number of visits to the cardiology clinic, number of health care services used, assessment of mental and physical well-being, presence of comorbidities, current symptoms, and nonadherence to physician recommendations. In addition, a higher HEHC score is strongly associated with the quality of life of family caregivers in the domains of body, mind, and social relationships, as well as with health-promoting behaviors and expectations for family caregivers.

### 4.2. Sociodemographics of Patients with CVD vs. Health Care Effectiveness

It should be noted that the incidence of health problems and chronic diseases increases with age. The number of CVD patients increases from the age of 60 years [19]. The self-report survey showed the relationship between age and EHC. It was found that respondents with lower EHC were older than those with potentially higher EHC (Me 77.5 vs. Me 70, *p* = 0.032). It should also be emphasized that logistic regression and odds ratio analyses revealed a much higher probability of HEHC in younger patients. This suggests that there is an urgent need to identify the health and social conditions and health and social needs of older people to improve the effectiveness of home care. Previous research has shown that socioeconomic conditions, such as education, income level, or living arrangements need to be seriously considered for a comprehensive understanding and appropriate assessment of CVD risk factors [31]. The level of income has also been shown to have an impact on increased morbidity and mortality from cardiovascular-related chronic diseases, such as diabetes and heart disease (OR = 1.36, 95% CI: 1.29–1.44 and OR = 1.24 (1.16–1.32). In addition, a financial barrier to care was associated with 30% higher inpatient costs [32]. The self-report study found that patients with potential LEHC had lower levels of education and financial status. The study on the criteria for hypertension and its complications showed that knowledge about this issue was much broader in patients with secondary or higher educational level than in those with primary educational level. The higher the level of education, the higher the level of knowledge [33]. In the current study, we showed that patients with higher educational level had a 1.45 (OR = 1.45, 95% CI: 1.05–2.02) times greater chance of HEHC and patients with better financial status had 2.31 (OR = 0.43, 95% CI: 0.21–0.83) times greater.

### 4.3. Clinical Data from CVD Patients vs. Health Care Effectiveness

The self-report study showed a strong correlation between EHC and the number of visits to a cardiology clinic (LEHC: Me—0.5 vs. HEHC: Me—2, *p* = 0.007). The results confirm previous findings by other investigators [34,35,36] emphasizing the positive impact of strengthening the role of the cardiologist within PHC. The self-reported study also found that LEHC correlated with health care services provided (LEHC: Me—0 (0.39) vs. HEHC: Me—0 (1.14), *p* = 0.032). The likelihood of possible HEHC increased with the frequency of procedures (OR = 1.25, 95% CI: 1.03–1.64). The occurrence of CVD has also been shown to be inversely correlated with good general health and satisfactory functioning [37]. However, the consequence of increased anxiety that a patient is unable to cope with in a situation of deteriorating health may lead to aggression [38]. The correlation between positive emotionality and longevity has also been confirmed. Good psychological well-being is associated with better immunological response. Moreover, optimism correlates significantly and independently with a lower number of indirect biomarkers of disease, as well as a lower incidence of pathologies and better and longer physical health [39]. Diseases limit patients in their daily activities and duties. They are accompanied by apathy, helplessness in the face of suffering, and fear of deteriorating health. The stresses and negative consequences on the physical and psychological state affect the quality of life and worsen the prognosis [39]. Evidence shows that noncompliance with medical recommendations is a serious problem for most patients and seems to be the main cause of unfavorable health outcomes. The phenomenon poses many dangers, especially in the management of chronic diseases [40]. The self-reported study found that LEHC was associated with low levels of aggression, negative self-assessment of one’s physical and mental well-being, as well as currently existing symptoms and irregular medication adherence. The coexistence of other conditions in patients with heart failure has also been found to be related to low levels of self-care (*p* < 0.05) [41] and to have a negative impact on quality of life and disability in chronically ill patients [42,43]. In the self-reported study, an association between EHC and the occurrence of concomitant diseases was found. The collected results suggest further research on this topic.

### 4.4. Variables Determining the Effectiveness of Health Care Compared to Caregivers of CVD Patients

Examining the correlates between EHC and CVD patients compared to their caregivers requires special attention in the discussion. Research shows that family and informal caregivers play an important role in home care. The role, situation, and needs of caregivers should be seriously considered when developing models of home care [17,20,44]. In the self-reported studies, caregivers are defined as individuals who usually have higher education and are therefore aware of the responsibilities and the range of possible activities that could slow down the development of a disease and positively affect a patient’s condition. It is worth noting that the correlation between EHC and nurses’ expectations was confirmed in the study (OR = 4.08, 95% CI: 1.31–14.33). The results clearly show that attention should be paid to these aspects among nurses. The most important issue among people who care for others is the social and emotional consequences. The sense of loneliness, isolation, and lack of information in this group of people leads to increased anxiety and fear of the future. In addition, caregivers report a lack of support with practical skills related to caring for an ill person [17,44,45]. The self-report study revealed an association between EHC and caregivers’ quality of life. The patients whose caregivers had high scores in the physical (OR = 1.24, 95% CI: 1.05–1.49), psychological (OR = 1.68, 95% CI: 1.25–2.37), and social (OR = 1.24, 95% CI: 1.09–1.44) domains had a greater chance of HEHC. Another important issue is the health-promoting behavior of caregivers. A caregiver caring for a chronically ill patient has the greatest impact on developing appropriate health-related attitudes. The study found that the overall intensity of caregiver health-promoting behaviors, which differed in intensity by 1 degree, may contribute to HEHC in patients. Patients whose caregivers scored higher on this item were 1.60 (OR = 1.60, 95% CI: 1.06–2.56) times more likely to have HEHC. The present data clearly show that the role of the caregiver, especially the family caregiver, has the greatest impact on the psyche (*p* < 0.01) [17]. The self-reported study proved that improvement in mental status after home care visits was strongly related to EHC. Patients whose caregivers reported low levels of positive mental attitude (the HBI) were 58.5 (OR = 0.02, 95% CI: 0.00–0.81) times more likely to have improved EHC than those for whom this score was high. However, the association between low levels of positive mental attitude according to the HBI and HEHC requires further research. The above characteristics indicate an attentive interest in this topic and a great need for professional support.

In summary, patient populations in primary care are heterogeneous in terms of clinical diagnoses. There are still too few such studies. Future research projects targeting heterogeneous patient groups in primary care should be considered.

### 4.5. Strengths of the Study

The strength of the study lies in the data analysis, which allowed defining a new variable-EHC- and dividing the patients into LEHC and HEHC. The study analyzed not only the differences between LEHC and HEHC patients, but also the correlations between EHC and the variables studied. In addition, the results can be used in the future to monitor changes in the effectiveness of home care (both nationally and internationally).

### 4.6. Limitations of the Study

There are several limitations to this study. The first crucial limitation of the study stems from the small sample group analyzed and conducting research in five voivodeships in Poland. This fact may significantly limit the possibility of generalizing the results to the overall population of CVD patients. It is worthwhile to study a larger group of patients and their caregivers, as well as a larger number of PHC clinics across the country. Second, the patient was included in the HEHC group when his results of WHOQOL-BREF), HBI, and the Camberwell index were higher than the corresponding values of 25% of the quantiles (first quartiles) of these three variables. Otherwise, the patient was included in the LEHC group. The non-complementarity of both groups results from the lack of data. The analogous criterion for the median (50% quantile) had a drawback: the number of one group was three times that of the other. Another limitation could be that, since we wanted to examine all possible models, we had to somehow limit the number of explanatory variables. The variables that were significantly correlated with the EHC variable at the 0.05 level were selected for logistic regression analysis because there were many more correlating variables at the 0.1 level. The next limitation could be low values of the pseudo-R^2^ indicator, which assesses the prediction of an explained variable using a model. The presented results of logistic regression analysis should be taken with caution because the pseudo coefficients of R-squared (coefficient of determination), which measure how well the model explains the collected data, are not large (0.23–0.39). Further research should seriously consider all of the above limitations.

### 4.7. Clinical Implications

Patients with cardiovascular disease and their caregivers can be well supported at home as long as the model of care is appropriate for them. It is necessary to conduct a pilot study to find the most effective and efficient model for integrating care for patients with chronic cardiovascular disease into primary health care. This includes coordination of family caregivers in the health care team, home care, and support from general practice. In addition, home care services should be monitored, regularly evaluated, and improved. Therefore, systems need to be created to identify at-risk groups with lower health care effectiveness among CVD patients in home PHC care. In addition, it is necessary to clinically inform and support decisions about personalized care models and provide clinical oversight and feedback to interdisciplinary teams about the patient’s condition. It is important to develop coordinated care plans, create educational programs for patients and caregivers, and develop policies useful for implementing system changes to identify individuals and support families caring for CVD patients. In addition, assess the knowledge and skills of health care workers in primary care settings who can participate in the delivery of home-based care.

## 5. Conclusions

Variables affecting the EHC of CVD patients included age, education, financial status, number of visits to cardiology clinics, number of health care services provided by a nurse, presence of comorbidities, medication irregularities, aggressiveness, self-assessment of physical and mental well-being, and improvement in mental well-being after a nursing home visit. The group of individuals with a potentially greater chance of HEHC includes patients whose caregivers report higher levels of physical, psychological, and social domains of quality of life, higher intensity of health-promoting behaviors, and lower ratings of positive mental attitude HBI, as well as expressing some expectation of caregivers.

## Figures and Tables

**Figure 1 ijerph-19-05170-f001:**
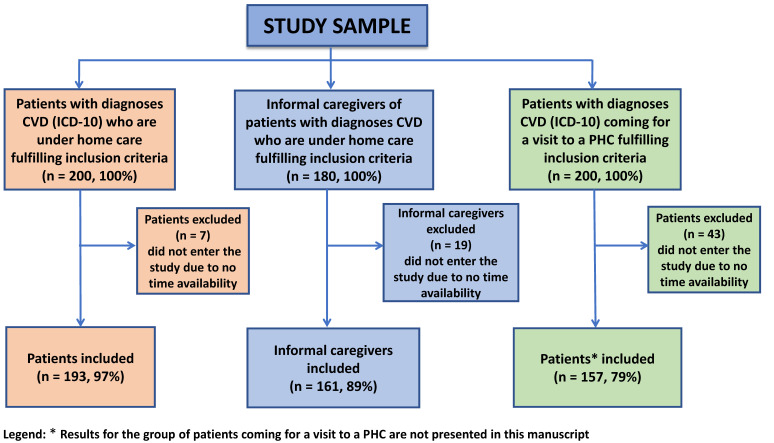
Flow diagram for the sample of CVD patients and their caregivers.

**Figure 2 ijerph-19-05170-f002:**
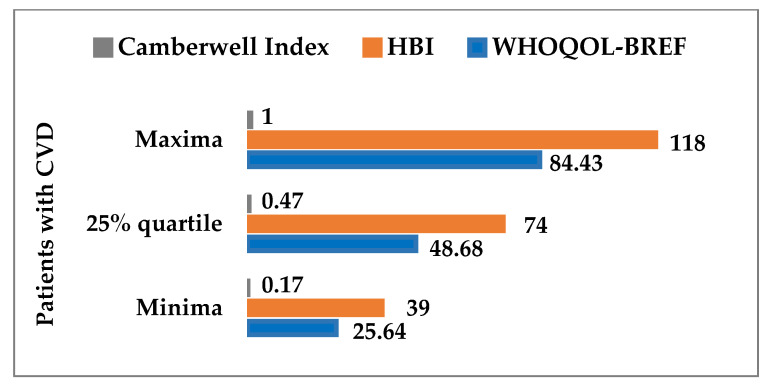
Better effectiveness of care for the group of patients under home care by a nurse—above the 25% quantile: WHOQOL-BREF quality of life, health behavior inventory, Camberwell index.

**Table 1 ijerph-19-05170-t001:** Clinical data of CVD patients with LEHC (*n* = 84 *) and HEHC (*n* = 85 *).

Variable		↓ LEHC				↑ HEHC				Wilcoxon Test	
		*n*	Me	q1	q3	*n*	Me	q1	q3	W	*p*
Duration of CVD (in years)		81	10	5	15	83	10	5	15	3471.5	0.718
Number of in the last 12 months	visits to PHC	84	5	1	10	85	5	3	10	3363.5	0.515
	cardiology clinic	84	0.5	0	2	85	2	0	2	2744.5	0.007
	home visits	84	2	0	4	85	0	0	3	3998	0.154
	telephone consultations	84	2	0	5.25	85	0	0	3	4111.5	0.068
	family nurse practitioner interventions	84	9.5	1	20.75	85	5	0	12	4013.5	0.16
Health care services (how many times/12 months)	medical interview	84	2	1	11.25	85	2	1	4	3781.5	0.501
	physical examination	84	4	0	12	85	3	0	6	3990	0.181
	blood pressure measurement	84	12	6	24.5	85	12	5	23	3757.5	0.556
	spirometry	84	0	0	0	85	0	0	0	3547	0.916
	diet control	84	3	1	6.5	85	2	0	7	3889.5	0.31
	BMI	84	1	0	2.25	85	0	0	2	3908	0.26
	pro-health education	84	6.5	2	12	85	6	1	12	3763	0.542
	others	84	0	0	0	85	0	0	0	3154.5	0.032
**Variable**	**Categories**	** *n* **	**%**			** *n* **	**%**			**Fisher test-*p***	
Current state of treatment (data provided by a nurse)	maintenance therapy	83	98.8			81	96.4			0.8	
	others	1	1.2			3	3.6				
	total	84	100			84	100				
Current state of treatment (data received from a patient)	maintenance therapy	79	96.3			79	95.2			0.073	
	no treatment	3	3.7			1	1.2				
	others	0	0			3	3.6				
	total	82	100			83	100				
Assessment of physical well-being	very bad	10	11.9			0	0			<0.001	
	bad	40	47.6			10	11.8				
	good	19	22.6			45	52.9				
	quite good	12	14.3			27	31.8				
	very good	3	3.6			3	3.5				
	total	84	100			85	100				
Assessment of mental well-being	very bad	6	7.2			0	0			<0.001	
	bad	35	42.2			7	8.3				
	good	20	24.1			50	59.5				
	quite good	20	24.1			18	21.4				
	very good	2	2.4			9	10.7				
	total	83	100			84	100				
Do you currently have any symptoms?	yes	53	63.1			36	42.9			0.011	
	no	31	36.9			48	57.1				
	total	84	100			84	100				
Adhere to the recommendations regarding lifestyle changes	yes	43	51.2			48	56.5			0.539	
	no	41	48.8			37	43.5				
	total	84	100			84	100				
Adhere to the recommendations regarding proper eating habits	yes	35	41.7			42	49.4			0.355	
	no	49	58.3			43	50.6				
	total	84	100			85	100				
Takes prescribed medications regularly	yes	54	64.3			73	85.9			0.001	
	no	30	35.7			12	14.1				
	total	84	100			85	100				

Legend: LEHC—patients with worse effectiveness of medical care; HEHC—patients with better effectiveness of medical care; *n*—group size; %—percentage; Me—median; q1 and q3—first and third quartiles; W—Wilcoxon test-*p* ≤ 0.05; Fischer test-*p* ≤ 0.05. * Numbers in column *n* do not sum to 84 and 85 due to missing data.

**Table 2 ijerph-19-05170-t002:** The results of the logistic regression analysis and the odds ratio of the logistic regression model in the group of urban residents. Explained variable: effectiveness of health care (0—if a patient belongs to a group with LEHC; 1—if a patient belongs to a group with HEHC).

Models with 8 Explanatory Variables			Odds Ratio						
**Model 1 (*n* = 130)**			**Per Unit**			**Per Range**			
Var.	Chi2 = 66.66, df = 8, *p* < 0.001, pseudo R^2^ = 0.37	b_i_	OR	95% CI	1/OR	OR	95% CI	1/OR	range
1	Age of patients (in years; 17–94)	−0.025	0.98	0.95–0.99	1.03	0.15	0.02–0.88	6.76	77
2	Difficulties in nursing care: not taking prescribed medications regularly (1—no, 2—yes)	−1.405	0.25	0.07–0.72	4.08	0.25	0.07–0.72	4.08	1
3	Attitude towards the disease and methods of treatment applied (1—positive, 2—negative)	−1.696	0.18	0.06–0.48	5.45	0.18	0.06–0.48	5.45	1
4	Financial status (1—very good, 5—very bad)	−0.836	0.43	0.21–0.83	2.31	0.08	0.01–0.58	12.26	3
5	HADS–M Aggression–patient (0—no, 6—high)	0.310	1.36	1.05–1.81	0.73	6.42	1.33–35.23	0.16	6
6	Patient: endocrinological disorders (1—no, 2—yes)	1.821	6.18	1.83–23.78	0.16	6.18	1.83–23.78	0.16	1
7	Improvement of a caregiver mental well-being after a nursing visit: I am full of hope and strength (1—no, 2—yes)	1.500	4.48	1.24–18.05	0.22	4.48	1.24–18.05	0.22	1
8	WHOQOL-BREF Social relations domain–caregiver (4—weak, 20—strong)	0.218	1.24	1.09–1.44	0.80	32.62	4.13–333.84	0.03	16
	**Model 2 (*n* = 130)**		**Per unit**			**Per range**			
Var.	Chi2 = 67.79, df = 8, *p* < 0.00001, pseudo R^2^ = 0.38	b_i_	OR	95% CI	1/OR	OR	95% CI	1/OR	range
9	Health care services (how many times/12 months) - others	0.226	1.25	1.03–1.64	0.80	29.60	1.48–1597.17	0.03	15
10	Education (1—primary, 7—post-secondary)	0.368	1.45	1.05–2.02	0.69	9.12	1.37–68.87	0.11	6
and	b_i_ values for the remaining variables in the model: (1) −1.519, (2) −1.912, (5) −0.787, (6) 0.346, (7) 1.411, (8) 0.190								
	**Model 3 (*n* = 120)**		**Per unit**			**Per range**			
Var.	Chi^2^ = 63.21, df = 8, *p* < 0.00001, pseudo R^2^ = 0.38	b_i_	OR	95% CI	1/OR	OR	95% CI	1/OR	range
11	Diagnosis of ICD-10: I99 (0—no, 1—yes)	−1.859	0.16	0.02–0.90	6.42	0.16	0.02–0.90	6.42	1
12	WHOQOL-BREF Psychological domain–caregiver (9.33—low, 18—high)	0.519	1.68	1.25–2.37	0.60	89.55	6.89–1768.19	0.01	8.67
13	HBI sten scale–caregiver (1—low, 10—high)	0.470	1.60	1.06–2.56	0.62	68.94	1.70–4648.61	0.02	9
14	HBI Proper mental attitudes–caregiver (2.33—weak, 5—strong)	−1.526	0.22	0.04–0.92	4.60	0.02	0.00–0.81	58.5	2.67
and	b_i_ values for the remaining variables in the model: (2) −1.764, (3) −2.914, (6) 1.494, (9) 0.573								
	**Model 4 (*n* = 124)**		**Per unit**			**Per range**			
Var.	Chi2 = 69.85, df = 8, *p* < 0.00001, pseudo R^2^ = 0.39	b_i_	OR	95% CI	1/OR	OR	95% CI	1/OR	range
15	Number of visits at cardiology clinic (within last 12 months) (0–24)	0.221	1.25	1.02–1.59	0.80	198.49	1.63–70,383.3	0.01	24
16	Carer’s expectations of higher manual skills while performing nursing duties towards a community nurse: (1—no, 2—yes)	1.405	4.08	1.31–14.34	0.25	4.08	1.31–14.33	0.25	1
17	WHOQOL-BREF Physical domain–caregiver (8.57—weak, 19.43—strong)	0.213	1.24	1.05–1.49	0.81	10.09	1.66–76.09	0.10	10.86
and	b_i_ values for the remaining variables in the model: (2) −1.416, (3) −1.983, (4) −1.294, (5) 0.308, (7) 1.689								
**Models with 7 Explanatory Variables**			**Odds Ratio**						
	**Model 5 (*n* = 124)**		**Per unit**			**Per range**			
Var.	Chi^2^ = 59.76, df = 7, *p* < 0.00001, pseudo R^2^ = 0.35	b_i_	OR	95% CI	1/OR	OR	95% CI	1/OR	range
18	Self-assessment of patient’s current mental well-being (1—very bad, 5—very good)	0.506	1.66	1.04–2.74	0.60	7.55	1.19–56.44	0.13	4
19	Nursing: endocrinological disorders (1—no, 2—yes)	1.240	3.46	1.08–12.35	0.29	3.46	1.08–12.35	0.29	1
and	b_i_ values for the remaining variables in the model: (3) −2.677, (4) −1.245, (11) −1.821, (17) 0.189, (18) 1.249								
	**Model 6 (*n* = 123)**		**Per unit**			**Per range**			
Var.	Chi^2^ = 38.63, df = 7, *p* < 0.00001, pseudo R^2^ = 0.23	b_i_	OR	95% CI	1/OR	OR	95% CI	1/OR	range
19	Self-assessment of patient’s physical well-being (1—very bad, 5—very good)	0.556	1.74	1.12–2.82	0.57	9.24	1.59–63.30	0.11	4
20	Nurse: urological disorders (1—no, 2—yes)	−1.824	0.16	0.04–0.55	6.20	0.16	0.04–0.55	6.20	1
and	b_i_ values for the remaining variables in the model: (7) 1.380, (11) −1.687, (13) 0.409, (14) −1.074, (15) 0.199								

Legend: Var.—variable designation, OR—odds ratio, CI-95% confidence interval for OR (range limits are given in brackets next to the description of the variable); Chi-squared—statistical hypothesis test of Chi2 model adjustment; df—number of degrees of freedom; *p*—calculated level of test significance; pseudo R^2^—value which evaluates explanatory variable anticipation according to the model; b_i_—estimating the coefficient beta in the logistic regression model; *n*—group quantity.

## Data Availability

The data presented in this study are available upon request from the corresponding author.

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
