# Peer review of "Variables Determining Higher Home Care Effectiveness in Patients with Chronic Cardiovascular Disease"

_ijerph, 2022, doi:10.3390/ijerph19095170_

Round 1
Reviewer 1 Report
Dear Editor and Authors,
It was a pleasure to evaluate the manuscript "Variables Determining Higher Level of Effectiveness of Home Care in Patients with a Chronic Cardiovascular Disease" of prof. Kurpas for publication in IJERPH. Thank you for your invitation.
The aim of this study was to was to find out wich factors influenced health care effectiveness of CVD patients in home care. The authors found that
The subject itself is interesting: out-of-hospital or primary care for chronic cardiovascular patients is infrequently studied, yet future improvement of cardiovascular care lie in these home care settings.
The conclusions are partially supported by the extensive data collected. There are serious reservations regarding novelty of the work including plagiarism, methodology, visualisation of the results, critical reflection and outlook.
I would like to comment as follows:
- There is tremendous overlap with recent papers in IJERPH and ethical approval identification number (Bioethical 488 Commission at Medical University in Wroclaw (No KB -86/2016) from the same group in the same period and patient group (doi:10.3390/ijerph17176427, doi:10.3390/ijerph18063231). It is unclear how the underlying study differs from the other studies, what the original protocol was about, and how findings and conclusions from both studies differ. It seems that a "salami-technique" is used to publish many papers around the same study group with only minor differences. Please explain.
- Furthermore, especially the methods section contains up to 100% overlap of sections with previously published papers as was detected in a Plagiarism check. Please explain
- The study design is unclear, and not described. The STROBE recommendations were not followed, i.e. primary and secondary endpoints were not described, and the STROBE checklist is lacking.
- There are ethical considerations regarding approval, obtaining informed consent of patients (response rate of 99%), potentially selecting variables. Also, an insane amount of questionnaires were filled in by patients. Can you estimate how much time was needed for these patients to fill in? For future submissions, include the original approved protocol including endpoints.
- Because the aim of the study is quite vague, design is not clear, and no figures were created, the overall flow of the manuscript is difficult to follow. It seems that choices should be made what to put in the main text, and what can be moved to supplements.
- It is unclear why the abstract mentions inversed odds ratios. furthermore, many and only significant findings were reported, thus giving a clear information bias to the readership of IJERPH. The abstract does not report implications of this study and future ideas to use the findings to improve home care for CVD patients
- The study group itself is inadequately described to make generalisation possible; how was CVD defined? Specify based on diagnosis, or treatment, or alternatively ICD-10 codes, as this might impact interpretation of the results
- The keywords have a 100% overlap with the title, and limits findability. Consider using synonyms, MeSH terms etc
- Introduction, lines 52-55: All patients (...) [5]. It is not only unlikely that all patients report this, but this statement is also not supported by REF 5
- Lines 57-59 are vague and potentially put the reader on the wrong ideas: the "large gap" was reported in low-and middle income countries (REF 7), and "all current activities" of whom focus on home enviroments of patients?
- The authors claim that this is the first study in Poland to look for factors influencing effectiveness in home care for CVD papers. In line with my comment 1 this is not only untrue, but information about other work in EU/globally is lacking.
- The sample size was not described. A study flow chart might improve readibility and flow of the manuscript to understand how the study was set up
- Methods, lines 98-99, why was a "partial analysis" done, 4 years after completion of the study? Why was the manuscript only now submitted? How many patients were in these voivodeships eligible for inclusion in this timeframe? Is "informal caregiver" the right term, if that group also included home care nurses? Would it be interesting to do a study with group-analysis between the type of caregivers?
- Section 2.4 is on the variables and data collection, which is clearly described. State in the introduction and/or discussion scores that were obtained in similiar patient groups in other groups to improve generalisation of your findings. Also consider to add further subsubheading per questionnaire to improve readibility
- Be clear: Did you ask for gender, or for sex? Change the manuscript to sex (most likely) including tables. Can you explain what a "material status" (line 175) is?
- Data analysis: The questionnaires have an ordinal scale and continous descriptions and/or tests were used, which is inaccurate. Please comment and adjust.
- There seems to be no justification for the group selection. Was there no possibility to use WHOQOL or HADS cut-off scores? Clinical relevance might therefore be limited. I also do not understand lines 197-199, and lines 202-209 seem to repeat these statements, but are slightly different.
- Make sure the tables are formatted in English and not Polish, and are easy to read. i.e. include range of each score in the "variable" cells (i.e. WHOQOL 24-120), state if mean (SD) or median (25-75 percentile) is presented etc.
- Describe how data is presented in section 2.5 i.e. mean +/- SD), median (25-75), and refrain from repeating that in results
- Why were 193 patients included in the study, but is table 2 only about 169 patients?
- Table 2 (and generally all tables): some units are missing, there is a mix of Polish and English words, inconsistent formatting of p-values, and generally a difficult to interpret format. Consider to move some tables or information to the supplements and make figures for improved visualisation
- Table 3: Inconsistent formatting. Unclear why categorical subvariables were tested individually, and not as Fisher exact.
i.e. if a variable is "number of home visits" with possible categories home visits, telephone consultations, etc, that should be analysed with 1 Fisher Exact test, and not on each subcategory. Many definitions are unclear or are not mentioned, consider to add that as a supplement to the manuscript - Section 3.4: the confidence intervals are lacking, please add. Furthermore, restructure and use subheadings for improved readibility. Consider to state only the main results in the results section, and move the rest to the supplements.
- There is only a very brief 4.6 limitations section without critical reflection. Please elaborate
- The 4.7 clinical implications section is quite brief and superficial. Please elaborate and insert a future outlook
- The references in discussion are outdated. Many Polish references are present. They can be interesting to show the validity of translated English to Polish scores, but are not accessible for 99% of the readership (not even the titles)
- The supplemental tables are an exact copy of the current tables in the manuscript.
Minor comments:
- Methods, lines 90-93, if you refer to the larger study, also refer to the (very recent) papers you published
- Lines 96-97 is it correct that caregivers only from the home care cohort were asked? Please clarify
- Lines 130-131 add a reference to the original WHOQOL questionnaire
- Line 214: why were variables with p < 0.05 included in the logistic regression, and not < 0.10? How were the final variables selected (line 217)?

Author Response
Dear Editor
International Journal of Environmental Research and Public Health
We would like to sincerely thank the Editorial Board and the Reviewer of your esteemed International Journal of Environmental Research and Public Health for their positive feedback and constructive recommendations to improve our observational paper entitled „Variables Determining Higher Level of Effectiveness of Home Care in Patients with a Chronic Cardiovascular Disease” by Elżbieta Szlenk-Czyczerska, Marika Guzek, Dorota Emilia Bielska, Anna Ławnik, Piotr Polański and Donata Kurpas.
In this first round of review, we focused heavily on the points raised in your letter. We would like to respond to this statement point by point based on our careful revision, as you can see in the table below. Accordingly, the final version of the manuscript text includes all necessary changes and improvements.
We sincerely hope that our revisions appear comprehensive and prove helpful in reaching a positive final decision on acceptance of our paper for publication in your prestigious International Journal of Environmental Research and Public Health.
Awaiting your decision, we remain yours faithfully
Authors

Reviewer 2 Report
This is an interesting topic that will be of interest to the readers of the journal. It presents a carefully designed cross-sectional statistical study aiming at identifying and analyzing the variables which may significantly affect the effectiveness of home care in patients with chronic cardiovascular diseases. It is generally well written and structured. The manuscript is clear and straight to the point. I have provided few minor revisions below:
Line 26: missing R in 1/OR
Line 28: statistical numerical figures for aggression are mistyped
The statistical results in the abstract need to be correctly presented; The CI values are not matched with the corresponding OR or 1/OR values. It is preferable to present all the odds ratios in the abstract either as OR or 1/OR per range.
Line 193: switch “better” and “worse”
It would be better to describe acronyms LEHC and HEHC at their first appearance.
Typos in table 1: maxsima
Data Analysis: merits rephrasing to be clearer
Table 2:
- Adjust column width to fit text
- total number of participants is 169, while it was stated earlier to be 191??
- Total number of patients in LEHC listed in table 2 title is 84, while in the table the sum would be 83.
- Typos: social
Remove space between lines 241 & 242
Lines 258-262: merits rephrasing as: “Mental status assessment showed that the number of patients with LEHC who reported their mental well-being as fairly good is equal to (LEHC: 24.1% vs 259 HEHC: 24.1%), as good (LEHC: 24.1% vs 260 HEHC: 59.5%) or very good (LEHC: 2.4% vs HEHC: 10.7%) less than, and as poor (LEHC: 42.2% vs HEHC: 8.3%) or very poor (LEHC: 7.2% vs HEHC: 0%) much more than that of patients with HEHC (p < 0.001).”
Line 253: Me not ME
Lines 255-256: Results stated are not mentioned in Table 3
Line 259: “fairly” changed to “quite”” as stated in Table 3
Line 259: % numbers of patients with LEHC or HEHC whose their mental well-being described as “fairly good” are mismatched with values listed in Table 3
Lines 262-263: change “poor” to “bad” as stated in Table 3
Line 266: mismatched numerical figures for patients with LEHC in line 266 and Table 3
Line 267: mismatched numerical figures for p value in line 267 and Table 3
Total numbers of CP with LEHC or HEHC listed within rows of Table 4 are greater than those listed in the title of Table 4
Kindly change material status to financial status wherever possible
The authors have chosen their models which comprise 8 statistically significant explanatory variables.
However:
In model 1, the variable “patient: endocrinological disorders”
AND
In model 4 & 5, the variables “Carer’s expectations of higher manual skills while performing nursing duties towards a community nurse” and “nursing: endocrinological disorders”
have not been screened and presented prior to logistic regression
Page 12: the numerical figures presented for variables 6-8 in model 3 in table 5 are not matching with those listed in lines 327-334.
Statistical numerical figures of I99 disease diagnosis (according to ICD-10) are not presented in table 5
Statistical numerical figures of the physical quality of life domain on the WHOQOL-BREF scale are not presented in table 5
Line 387: typos analyses
Line 399-400: two different variables have been compared?
It would be better to use “in the current, present….study” instead of “in the self-report study” so as not confuse and mislead the readers
Apparently, the present results regarding correlation between levels of aggression and LEHC are not in line with those reported in reference 30. May the authors generously offer an explanation for such contradictory results?
Author Response

(The authors gave the same response as above.)

Reviewer 3 Report
Czyczerska et al., revealed that, variability that influence the effectiveness of home care on health behaviours in patients with cardiovascular diseases. Very interesting study, as per the Sociodemographic data shows number of women patients were higher then the men with cardiovascular diseases. further the age group is above 75. Authors need to consider below 75 years in their future studies. in addition, authors need to consider revising the data tables with charts to understand better.
Author Response

(The authors gave the same response as above.)

Reviewer 4 Report
The manuscript provides an interesting insight into in the effectiveness of home care among CVD patients. However, it might be expected that people with better health care effectiveness would be younger, with better mental and physical condition, better educated, have better material status, make more frequent visits to the cardiology clinic, use health services, and follow medical recommendations.
Some issues need to be addressed to improve the quality and clarity of the manuscript for publication. Specific comments are provided below for consideration:
The abstract is far too long and contains an overwhelming amount of results.
The introduction is written too generally and contains a lot of biased sentences. Perhaps more statistical data on the subject would be useful. Have such studies been conducted outside of Poland? Has any country introduced organized patient home care and how did it work? How did it improve the budget?
Results should be more well presented. The tables are unreadable and poorly organized. The authors might consider a clearer way of presenting the results?
Please pay attention to the data in Table 2. After adding up the number of men and women we do not get the whole number of patients in the LEHC group? A similar situation occurs in Table 4.
In what units is the ,,Duration of CVD illness'' in Table 2 defined?
Are you sure that only patients who were eighteen years old could be included in the study? And not older ones?
Author Response
Dear Editor
International Journal of Environmental Research and Public Health
We would like to sincerely thank the Editorial Board and the Reviewer of your esteemed International Journal of Environmental Research and Public Health for their positive feedback and constructive recommendations to improve our observational paper entitled „Variables Determining Higher Home Care Effectiveness in Patients with Chronic Cardiovascular Disease” by Elżbieta Szlenk-Czyczerska, Marika Guzek, Dorota Emilia Bielska, Anna Ławnik, Piotr Polański and Donata Kurpas.
In this first round of review, we focused heavily on the points raised in your letter. We would like to respond to this statement point by point based on our careful revision, as you can see in the table below. Accordingly, the final version of the manuscript text includes all necessary changes and improvements.
We sincerely hope that our revisions appear comprehensive and prove helpful in reaching a positive final decision on acceptance of our paper for publication in your prestigious International Journal of Environmental Research and Public Health.
Awaiting your decision, we remain yours faithfully
Authors

Round 2
Reviewer 1 Report
Dear Editor and Authors,
It was a pleasure to evaluate the revised manuscript (R1) "Variables Determining Higher Home Care Effectiveness in Patients with Chronic Cardiovascular Disease" for publication in IJERPH. Thank you for your invitation.
The authors have put effort in revision of the original manuscript based on the reviewers' comments, applied the STROBE framework to their manuscript, now used subheadings for improved readibility, and included a new figure. Their well-formatted point-to-point response is appreciated.
Major initial concerns related to plagiarism, original content, and ethics were adequately addressed.
Unfortunately, many comments were only partly addressed. The overall manuscript remains lengthy with 18 pages, with 5 lengthy and complex tables, and one new (and potentially incorrect) figure. The manuscript remains difficult to read. The authors did not choose to select the most important findings, and transfer the rest to supplementary material, nor did they choose to include figures to improve readibility.
Original comments
Comments 1-2, 8-10, 16, 26-27 were all addressed and improved the manuscript, thank you.
3. Thank you for using the STROBE recommendations. This improved the manuscript. Some STROBE-items are not traceble in the manuscript. i.e. the checklist reports that study size was mentioned on page 5, but this is lacking. Make sure to address all items in the manuscript
4. This comment on ethical considerations was partly addressed. What is your response rate? 89% and 97%? Please comment. Also see the comment on the flow chart.
5. The design is now clear, figure 1 improved the manuscript (see further comment on flow chart), yet the latter comment was not addressed: Make choices what to put in the main text, and what can be moved to supplements. Also, the aim at the introduction is still somewhat vague and this comment was not addressed.
6. My comments on inversed odd ratios, reporting of only significant values resulting to bias were not addressed.
7. There is now more text, but not the information I was looking for. How was cardiovascular disease defined? This is crucial for understanding the patient population.
11. Thank you for refering to the Polish studies from your group and the other EU and USA study. Why is it problematic that these studies focus on homogeneous groups only? Please comment, also in the manscript discussion.
12. My comment on sample size was not addressed, see comment 3
13. Thank you for adressing most points. One comment was not addressed: how many patients were in these voivodeships eligible for inclusion in this timeframe?
14. The subheadings improved readibility of the manuscript, thank you. The scores were not reported, therefore, it is impossible to compare your work to others.
15. Thank you for changing gender to sex. Change "material status" into "financial status" also in line 181, 398
17. Your statement "The analogous criterion in patients cared for by a nurse at home based on the median (50% quantile) had a disadvantage: the figure for one group was three times that of the other." is not clear to me. Can you please rephrase?
18. This comment was partly addressed. School system not explained, making generalisation difficult.
19. Only presenting median is insufficient. Please also report 25 and 75 interquartile range
20. This is still not clear for me. How is it possible that you discard 50% of the data (because you only select the top 25% (HEHC) and lowest 25% (LEHC), but 169/193=88% is included in Table 2?
21. Improved to some extent; it is more consistent. According to the IJERPH template, table should be formatted full page width (MDPI 4.2 Table body). It is still cramped and not easy to understand/read
22. The first comments were adressed; this is clear. The definitions are still lacking. Consider to include them in the supplements
23. This was partly addressed. See comment 5
24. Thank you for improving the limitations section.
25. The changes did not improve the manuscript. i.e. one sentence is 73 words. Please rephrase.
Original minor comments
Minor comments 1, 3-4 were all addressed and improved the manuscript, thank you.
2. Thank you for explaining. The information presented here is missing from the flow chart, and also, this illustrates that the flow chart is incorrect I think?
My understanding is: First you start with 193+7+157 patients
> 157 are not included because they went to their GP
Then 193+7 patients were in this study.
> 7 excluded because of time
Of these 193 patients, 161+19 of their informal caregivers participated.
> 19 were exluded because of time
If this is correct, please change the flow chart.
New comments (R1)
- For reviewing purposes, please include line numbers for all questions, subquestions and changes to the manuscript.
- Abstract, line 34: be clear, are respondents the patients, informal caregivers, or both?
- Abstract, line 38: "tailored to support them". Tailored to what? Please specify
- Methods, line 105-106: It is great to see that the family nurse encouraged patients and their informal caregivers to participate, resulting in 97% and 89% response rates (If I am correct). You can opt to mention these percentages in the manuscript to illustrate this high response rate
- Figure 1, first see original minor comment 2. Also: "180 caregivers" should be "180 informal caregivers"?
- Figure 1, If created in Powerpoint, consider to export to PDF first, to maintain vector standard and therefore improve figure quality.
- Figure 1, two typos: "due to not time availability" should be "due to no time availability"
- Methods, lines 128-129: how can this study be anonymous if the family nurse is present? What is the influence of a family nurse being present? Please comment on both aspects.
- Results, line 250: "and fewer dit not..." is redundant, because the opposite is already mentioned in that sentence. I suggest to leave that out to improve readibility
- Table 2, How can social benefit (data from a nurse) be scored if it was anonymous?
- Section 3.2. subheading I think a word is missing after "clincial"
- Results, lines 262-267 are important for studying healthcare inequalities. Come back to this in the discussion.
- The ables, i.e. Table 3 and on are still difficult to follow. Some suggestions: Table formatting ideas, see:
Int. J. Environ. Res. Public Health 2022,
19, 4329. https://doi.org/10.3390/
ijerph19074329
(recent paper in IJERPH)There are also box-and-whisker plots in that paper to illustrate results. I.e. try to convert Table 1 into a figure.
Also i.e. 2nd part of table 3 into figure, and 2nd part table 4 into figure
- Limitations, lines 494-498 Is the study size really small?
Author Response
Dear Editor
International Journal of Environmental Research and Public Health
We would like to sincerely thank the Editorial Board and the Reviewer of your esteemed International Journal of Environmental Research and Public Health for their positive feedback and constructive recommendations to improve our observational paper entitled „ Variables Determining Higher Home Care Effectiveness in Patients with Chronic Cardiovascular Disease” by Elżbieta Szlenk-Czyczerska, Marika Guzek, Dorota Emilia Bielska, Anna Ławnik, Piotr Polański and Donata Kurpas.
In this second round of review, we have focused heavily on the points raised in your letter. We would like to respond to the comments point by point, as you can see in the table below. Accordingly, the final version of the manuscript text includes all necessary changes and improvements.
We sincerely hope that our revisions appear comprehensive and prove helpful in reaching a positive final decision on acceptance of our paper for publication in your prestigious International Journal of Environmental Research and Public Health.
Awaiting your decision, we remain yours faithfully,
Authors

Reviewer 4 Report
After correction of manuscript by Authors I accept the paper for publishing.
Author Response
Dear Editor
International Journal of Environmental Research and Public Health
We would like to sincerely thank the Editorial Board and the Reviewer of your esteemed International Journal of Environmental Research and Public Health for their positive feedback and constructive recommendations to improve our observational paper entitled „Variables Determining Higher Home Care Effectiveness in Patients with Chronic Cardiovascular Disease” by Elżbieta Szlenk-Czyczerska, Marika Guzek, Dorota Emilia Bielska, Anna Ławnik, Piotr Polański and Donata Kurpas.
In this second round of review, the authors thank the reviewer for the positive feedback.
Authors
